# Sampling-Enhanced Large Neighborhood Search for Solving Integer Linear Programs

## Abstract

Large Neighborhood Search (LNS) is a common heuristic in combinatorial optimization that iteratively searches over a large neighborhood of the current solution for a better one. Recently, neural network-based LNS solvers have achieved great success in solving Integer Linear Program (ILP) problems with a learnable policy for neighborhood selection, followed by an off-the-shelf ILP solver for re-optimization. Nonetheless, existing neural LNS solvers often get stuck in the same solution due to their greedy update strategy, i.e., only moving to the best solution found within the neighborhood. In this work, we try to theoretically identify the limitation of neural models in escaping the "local optima". Accordingly, we propose a novel sampling-enhanced neural LNS solver, namely SPL-LNS, by reformulating LNS as a stochastic process, which uses a locally-informed proposal to sample the next assignment and simulated annealing to alleviate the "local optima" issue. We also develop a novel hindsight relabeling method to efficiently train SPL-LNS on self-generated data. Experimental results reveal that our method substantially surpasses prior neural LNS solvers on multiple ILP problems.

## 1 Introduction

Combinatorial Optimization (CO) problems present a set of fundamental challenges in computer science for decades (Papadimitriou and Steiglitz, 1998). Many of those problems can be formulated as generic Integer Linear Programs (ILPs), including supply chain management, logistics optimization (Chopra and Meindl, 2001), workforce scheduling (Ernst et al., 2004), financial portfolios (Mashayekh et al., 2017), compiler optimization (Trofin et al., 2021; Zheng et al., 2022), bioinformatic problems (Gusfield, 1997), and more. Classic ILP solvers typically conduct a tree-style search with the Branch-and-Bound (BnB) algorithm (Land and Doig, 2010), which finds the *exact* solution by gradually reducing and finally closing the gap between the primal (upper) and dual (lower) bounds of the searched solutions. Many state-of-the-art open-source and commercial ILP solvers are of this kind, including SCIP (Achterberg, 2009), CPLEX (Cplex, 2009), and Gurobi (Gurobi Optimization, 2021). However, when the problems are very large, completely closing the primal-dual gap can be intractable. Hence, solvers for large ILP problems have shifted efforts towards *primal heuristics* (Berthold, 2006), which are designed for finding the best possible solutions within a limited time window. That is, those heuristic primal solvers do not guarantee to find the optimal solutions, but aim to tackle large ILP problems with near-optimal solutions. Our work in this paper belongs to the category of heuristic-driven primal solvers.

Large Neighborhood Search (LNS) is a typical heuristic strategy for finding high-quality solutions much faster than pure BnB (Ahuja et al., 2002). The process starts with a poor solution and iteratively revises the current solution with the *destroy* and *repair* operations. In each iteration, the system selects (or destroys) a subset of the variables, then re-optimizes (or repairs) them while keeping the other variables unchanged. Finding good heuristics for the destroying and repairing operators has been a central focus of the LNS-based solvers for ILP. Recently, neural network-based LNS methods have shown great potential because they can learn from vast amounts of data. For example, Song et al. (2020b) and Wu et al. (2021) presented a general neural LNS pipeline where the destroy operator was parameterized with a learning-based neural policy, and the repair operator was carried out by an off-the-shelf ILP solver. To obtain high-quality supervision, Sonnerat et al. (2021) and Huang et al. (2023b) utilized a powerful expert heuristic called local branching (LB) (Fischetti and Lodi, 2003b), which guarantees to find the locally optimal destroy policy given enough solving

time. When the neighborhood size is sufficiently large, LNS is believed to be free from the issue of getting stuck in poor local optima (Sonnerat et al., 2021). Therefore, the greedy update with the best-found solution in the neighborhood has been a common strategy in existing neural LNS solvers.

However, we have found that in practice, such a greedy update strategy often traps the system in a "local optima", in the sense that the chance for a destroy operator to find a suitable destroyed set leading to a better solution becomes extremely small, and the search would always stay around the same solution. How to overcome such a difficulty remains an open challenge for neural LNS solvers for ILPs. In this paper, we propose a novel SamPLing-enhanced neural LNS method, namely SPL-LNS, which draws a connection between LNS and the discrete Markov Chain Monte Carlo (MCMC) with a locally-informed proposal (Zanella, 2017). By reformulating LNS as a stochastic process, we tackle the "local optima" issue with simulated annealing (Kirkpatrick et al., 1983), which has been proven to be effective in addressing local optima in other CO problems (Johnson et al., 1991; Sun et al., 2023). The main difference in our proposed approach is that we sample the next assignment from a set of feasible solutions within the neighborhood rather than greedily pick the best one. We also develop a novel hindsight relabelling strategy to efficiently generate informative training data for our sampling-enhanced neural LNS. Specifically, we use a trained neural network to collect the LNS trajectories during the search process, and relabel the optimal destroyed variables for each sample in the trajectory to obtain high-quality supervision. These self-generated training data, combined with the supervision from the expert heuristic, can be used to train a more powerful neural destroy policy under our sampling-enhanced LNS framework.

To summarize, our work tries to address a fundamental question in Combinatorial Optimization about what can be learnt through neural models and what cannot. Our theoretical analysis indicates that neural models could provide a strong heuristic in proposing the potential region containing high-quality solutions, but they lack the ability to escape the "local optima". Therefore, we tackle this limitation via a sampling-enhanced model. Our empirical evaluation shows that SPL-LNS consistently outperforms the state-of-the-art neural LNS solvers and traditional heuristic methods on four synthetic ILP benchmarks and a real-world ILP problem in multiple metrics.

## 2 PRELIMINARIES

**Integer Linear Program**   ILP is a type of discrete optimization problem whose variables are subject to integrality constraints. The general form of an ILP problem could be expressed as

$$
\begin{aligned}
\min \mathbf{c}^\top \mathbf{x} & \\
\text{s.t. } \mathbf{A}\mathbf{x} \leq \mathbf{b}, \ \mathbf{x} \in \mathbb{Z}^n,
\end{aligned}
\tag{1}
$$

where $\mathbf{x} = (\mathbf{x}_1, \cdots, \mathbf{x}_n)^\top$ is the vector of decision variables, $\mathbf{c} \in \mathbb{R}^n$ is the vector of objective coefficients, $\mathbf{A} \in \mathbb{R}^{m \times n}$ and $\mathbf{b} \in \mathbb{R}^n$ represent the constraint coefficients. In the following context, all functions are conditioned on the input ILP and we no longer write it out explicitly.

Some common metrics in measuring the solution quality to an ILP includes (1) *primal bound*: the objective value $\mathbf{c}^\top \mathbf{x}$ for the incumbent solution $\mathbf{x}$; (2) *primal gap* (Berthold, 2006): the normalized difference between the primal bound and a pre-computed optimal (or best known) objective value $\mathbf{c}^\top \mathbf{x}^*$, defined as $\frac{|\mathbf{c}^\top \mathbf{x}^* - \mathbf{c}^\top \mathbf{x}|}{\max\{|\mathbf{c}^\top \mathbf{x}^*|, |\mathbf{c}^\top \mathbf{x}|\}}$ when the solution $\mathbf{x}$ is found and $\mathbf{c}^\top \mathbf{x}^* \cdot \mathbf{c}^\top \mathbf{x} \geq 0$, or 1 otherwise; (3) *primal integral* (Berthold, 2006): the integral of the primal gap on the time range $[0, T]$.

**Large Neighborhood Search**   LNS is a process for iteratively improving the solution by the *destroy* and *repair* operators. It starts with an initial feasible solution $\mathbf{x}^{(0)}$, typically obtained by running a traditional symbolic solver with a limited time budget. In its $t$-th iteration, the destroy operator heuristically chooses a subset of the decision variables in the current solution $\mathbf{x}^{(t)}$ and the repair operator re-optimizes the next solution $\mathbf{x}^{(t+1)}$ over the destroyed variables while keeping the remainder unchanged. Here we use $\mathbf{d}^{(t)} \in \{0, 1\}^n$ to denote the destroyed set where 1 indicates destroyed. $\eta^{(t)} = \|\mathbf{d}^{(t)}\|_1$ is the neighborhood size at $t$-th iteration. The re-optimization step is typically carried out by an off-the-shelf ILP solver that could always attain the best solution inside the preset neighborhood with enough solving time. In practice, an adaptive neighborhood size is typically used to control the difficulty of the re-optimization step (Sonnerat et al., 2021; Huang et al.,

2023b). For example, Huang et al. (2023b) increase the neighborhood size $\eta^{(t+1)} = \min\{\gamma\eta^{(t)}, \beta n\}$ when no improved solution is found, where $\gamma \geq 1$ is a constant and $\beta < 1$ controls the upper bound.

Since a large neighborhood size almost surely guarantees the existence of a better assignment before LNS reaches a sufficiently good one, existing neural LNS solvers always update the solution to the best one found within the neighborhood during each iteration. Most research efforts in neural LNS have thus focused on developing effective heuristics for the destroying operator to speed up the overall process and largely ignored the effectiveness of such a greedy update strategy in practice.

**Local Branching** LB (Fischetti and Lodi, 2003a) is first used by Sonnerat et al. (2021) as an expert destroy heuristic to generate the supervision for neural LNS. It formulates the optimal neighborhood selection for LNS as another ILP and searches for the next optimal solution $\mathbf{x}^{(t+1)}$ inside the Hamming ball with radius of $\eta^{(t)}$ centered around the current solution $\mathbf{x}^{(t)}$. LB obtains the locally optimal destroy policy that provides informative supervision. However, LB owns the same computational complexity as the original ILP and as LNS proceeds, its astronomical solving time would become unaffordable even for the data collection purpose. Therefore, as a locally optimal policy, LB would also get stuck once it fails to find a better solution.

**Locally-Informed Proposals** Locally-informed proposals are developed by Zanella (2017) for efficient Markov Chain Monte Carlo (MCMC) algorithms. Consider a target distribution $p(x) = \exp\left(-\frac{1}{\tau}E(x)\right)/Z$ defined over $\mathbf{x} \in \mathbb{Z}^n$. Here $E(\mathbf{x})$ is an energy function, $Z = \sum_{\mathbf{x}'} \exp\left(-\frac{1}{\tau}E(x')\right)$ stands for the normalizing constant which is assumed to be intractable, and $\tau > 0$ represents the temperature. Metropolis-Hastings (MH) (Metropolis et al., 1953; Hastings, 1970) algorithm provides a generic MCMC framework to sample from such a distribution with the rejection sampling. In each iteration, it samples $\mathbf{x}'$ from a proposal distribution $q(\mathbf{x}'|\mathbf{x}^{(t)})$ given the current state $\mathbf{x}^{(t)}$, then with probability $\min\{1, \frac{p(\mathbf{x}')q(\mathbf{x}^{(t)}|(\mathbf{x}'))}{p(\mathbf{x}^{(t)})q(\mathbf{x}'|\mathbf{x}^{(t)})}\}$, $\mathbf{x}'$ would be accepted as $\mathbf{x}^{(t+1)}$, otherwise $\mathbf{x}^{(t+1)} = \mathbf{x}^{(t)}$. Such an iterative process forms a Markov Chain $\mathbf{x}^{(0)}, \mathbf{x}^{(1)}, \cdots$ with the stationary distribution $p(\mathbf{x})$.

A good proposal distribution needs to balance $p(\mathbf{x}')$ and $q(\mathbf{x}^{(t)}|\mathbf{x}')$ to achieve a high acceptance rate and fast convergence. One kind of these proposals is the locally-informed proposal (Zanella, 2017)

$$q(\mathbf{x}'|\mathbf{x}) \propto g(p(\mathbf{x}')/p(\mathbf{x}))K(\mathbf{x}', \mathbf{x}), \tag{2}$$

where $g$ is a scalar weight function (e.g., $g(y) = \sqrt{y}$ or $\frac{u}{y+1}$), and $K$ is a symmetric kernel with the same density for $K(\mathbf{x}', \mathbf{x})$ and $K(\mathbf{x}, \mathbf{x}')$. The locally-informed proposals have also recently been found useful in other combinatorial optimization problems (Sun et al., 2023) when equipped with simulated annealing (Kirkpatrick et al., 1983), which gradually anneals $\tau$ towards 0, allowing the search to escape local optima and reach a low-energy state with high probability.

## 3 METHOD

Our approach is motivated by the observation that LNS would get stuck when the destroy operator fails to find the suitable destroy set for a better solution despite its existence. We start with an analysis of the universal nature of such a problem and then introduce our proposed method SPL-LNS in detail.

### 3.1 UNDERSTANDING THE "LOCAL OPTIMA" IN LNS

LNS is believed to be less susceptible to the local optima problem due to the large search space, and it could always attain a better solution with a perfect destroy heuristic. However, such a perfect heuristic does not exist in practice and LNS would still suffer from the "local optima" problem when the chance for a destroy heuristic to find a suitable destroyed set dramatically decreases.

Denote the new ILP to be solved by the repair operator at $t$-th iteration as $\mathbf{P}^{(t)} = P(\mathbf{d}^{(t)}, \mathbf{x}^{(t)})$, which is the ILP in Equation 1 with additional constraints $\{\mathbf{x}_i = \mathbf{x}_i^{(t)}|i : \mathbf{d}_i^{(t)} = 0\}$. Represent the objective value for any feasible solution as a random variable $u = \mathbf{c}^\top \mathbf{x}$ with respect to $\mathbf{d}^{(t)}$. Without

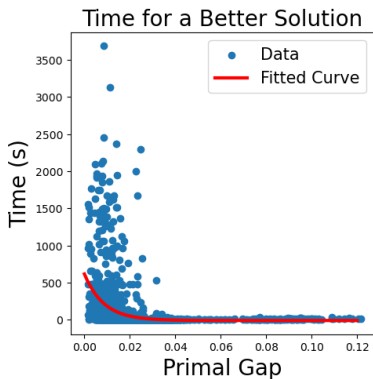

Figure 1: Exponentially increasing time to find a better solution as LNS proceeds (primal gap $\rightarrow$ 0). The statistics are from the Combinatorial Auction problem, see Section 4.1 for details.

---

**Algorithm 1** SPL-LNS

1: **Input**: An ILP with $(\mathbf{A}, \mathbf{b}, \mathbf{c})$
2: Find an initial feasible solution $\mathbf{x}$
3: Initialize $\mathbf{x}^* \leftarrow \mathbf{x}$, $\tau$ and $\sigma$
4: **while** stopping criterion is not met **do**
5:    Sample the destroyed variables $\mathbf{d}$ with $\pi_\theta$
6:    Create the sub-ILP $\mathbf{P} \leftarrow P(\mathbf{x}, \mathbf{d})$
7:    Solve $\mathbf{P}$ to obtain feasible solutions $\mathcal{S}(\mathbf{P})$
8:    **if** $\min_{\mathbf{x}' \in \mathcal{S}(\mathbf{P})} \mathbf{c}^\top \mathbf{x}' < \mathbf{c}^\top \mathbf{x}^*$ **then**
9:       $\mathbf{x}^* \leftarrow \arg\min_{\mathbf{x}' \in \mathcal{S}(\mathbf{P})} \mathbf{c}^\top \mathbf{x}'$
10:   **end if**
11:   Sample $\mathbf{x}'$ from $\mathcal{S}(\mathbf{P})$ with $p_r$
12:   **if** Accept($\mathbf{x}'$) **then**
13:      $\mathbf{x} \leftarrow \mathbf{x}'$
14:   **end if**
15:   Update $\tau$ and $\sigma$
16: **end while**
17: **return** $\mathbf{x}^*$

---

loss of generality, assume $u$ is always negative. Represent the chance for a repair operator to find a solution not worse than the current one as $p(u \le \mathbf{c}^\top \mathbf{x}^{(t)})$, then by Chernoff bound,

$$p(u \le \mathbf{c}^\top \mathbf{x}^{(t)}) \le \frac{\mathbb{E}[\exp(su)]}{\exp(s\mathbf{c}^\top \mathbf{x}^{(t)})}, \ \forall s < 0. \tag{3}$$

Here, $\mathbb{E}[\exp(su)]$ is computed over the set of all feasible solutions, denoted as $\mathcal{S}(\mathbf{P}^{(t)})$, for the sub-ILP $\mathbf{P}^{(t)}$. Let us consider two extreme cases. When $\eta^{(t)} \rightarrow 0$, $\mathbb{E}[\exp(su)] \rightarrow \exp(s\mathbf{c}^\top \mathbf{x}^{(t)})$. However, unless there happens to be a better solution within the small neighborhood (only happens at the initial stage of LNS), solving $\mathbf{P}^{(t)}$ would only recover $\mathbf{x}^{(t)}$ itself as the best solution. When $\eta^{(t)} \rightarrow n$, $\mathbb{E}[\exp(su)]$ would almost become a constant (an expectation over all feasible solutions to the original ILP), and $p(u \le \mathbf{c}^\top \mathbf{x}^{(t)})$ would drop exponentially as the solving proceeds. In LNS, a large neighborhood size makes $\mathbb{E}[\exp(su)]$ closer to the second case, especially in the later stage when better solutions are far away from the current solution. Consequently, the time to find a better solution would also increase exponentially, as we visualized in Figure 1 for solving LNS on the Combinatorial Auction dataset. Therefore, although an adaptive neighborhood size (Sonnerat et al., 2021; Huang et al., 2023a;b) could help escape the local optima (meaning no better solutions within the Hamming ball/neighborhood) by expanding the neighborhood size, LNS would still get stuck in the "local optima" in the sense that the chance to find a suitable destroy set and thus the better solution becomes too small. Hence, a more efficient update strategy to avoid wasting the search around the same solution is needed to tackle this "local optima" problem.

### 3.2 SAMPLING-ENHANCED LARGE NEIGHBORHOOD SEARCH

The key idea of our sampling-enhanced Large Neighborhood Search (SPL-LNS) method is to overcome such a "local optima" in LNS via the well-known simulated annealing approach (Kirkpatrick et al., 1983). Our first step is to formulate LNS as a stochastic process as in MCMC. Although LNS and locally-informed proposals share high similarities, no prior work has systematically bridged the gap between them. We now aim to make the connection by treating the destroy and repair operators as two distinct distributions to sample from, denoted as $p_d(\cdot|\mathbf{x})$ and $p_r(\cdot|\mathbf{d}, \mathbf{x})$. Hence, each update in LNS could be treated as sampling $\mathbf{x}^{(t+1)}$ from the following distribution

$$p(\mathbf{x}^{(t+1)}|\mathbf{x}^{(t)}) = \sum_{\|\mathbf{d}\|_1 = \eta^{(t)}} p_r(\mathbf{x}^{(t+1)}|\mathbf{d}, \mathbf{x}^{(t)}) p_d(\mathbf{d}|\mathbf{x}^{(t)}). \tag{4}$$

Consider a target distribution with the energy function

$$E(\mathbf{x}) = \begin{cases} \mathbf{c}^\top \mathbf{x}, & \text{if } \mathbf{A}\mathbf{x} \le \mathbf{b}, \\ +\infty, & \text{otherwise.} \end{cases} \tag{5}$$

Then LNS could be treated as an MCMC process converging to $p(x) = \exp(-E(x)/\tau)/Z$ with a small $\tau$. To accelerate the convergence of this stochastic process to the target distribution, we can make the proposal distribution $p(\mathbf{x}^{(t+1)}|\mathbf{x}^{(t)})$ as the following locally-informed proposal

$$p(\mathbf{x}^{(t+1)}|\mathbf{x}^{(t)}) \propto \exp\left(\frac{1}{2\tau}(E(\mathbf{x}^{(t)}) - E(\mathbf{x}^{(t+1)}))\right)\binom{n - d_H(\mathbf{x}^{(t+1)}, \mathbf{x}^{(t)})}{\eta^{(t)} - d_H(\mathbf{x}^{(t+1)}, \mathbf{x}^{(t)})}, \qquad (6)$$

where $d_H(\cdot, \cdot)$ measures the Hamming distance between two inputs. The above formulation would yield the following two distributions for the destroy and repair operators

$$p_d(\mathbf{d}|\mathbf{x}^{(t)}) = \frac{\sum_{\mathbf{x}' \in \mathcal{S}(P(\mathbf{d}, \mathbf{x}^{(t)}))} \exp\left(-\frac{1}{2\tau}E(\mathbf{x}')\right)}{\sum_{\|\mathbf{d}'\|_1 = \eta^{(t)}} \sum_{\mathbf{x}' \in \mathcal{S}(P(\mathbf{d}', \mathbf{x}^{(t)}))} \exp\left(-\frac{1}{2\tau}E(\mathbf{x}')\right)}, \qquad (7)$$

$$p_r(\mathbf{x}^{(t+1)}|\mathbf{d}, \mathbf{x}^{(t)}) = \frac{\exp\left(-\frac{1}{2\tau}E(\mathbf{x}^{(t+1)})\right)\mathbb{1}[\mathbf{x}^{(t+1)} \in \mathcal{S}(P(\mathbf{d}, \mathbf{x}^{(t)}))]}{\sum_{\mathbf{x}' \in \mathcal{S}(P(\mathbf{d}, \mathbf{x}^{(t)}))} \exp\left(-\frac{1}{2\tau}E(\mathbf{x}')\right)}. \qquad (8)$$

We include the full derivation in Appendix A. Equation 8 is a distribution defined by the energy-based model over all feasible solutions to the sub-ILP $P(\mathbf{d}, \mathbf{x}^{(t)})$, and we could easily model it using the feasible solutions found by the ILP solver. In our implementation, we use the top-$k$ feasible solutions found by the ILP solver as the candidate set to draw the sample and make the rejection sampling an optional choice due to the high acceptance rate. Here $k$ is a hyperparameter.

Equation 7 is instead intractable due to the combinatorial nature of the destroyed set. Here we note that $p_d(\mathbf{d}|\mathbf{x}^{(t+1)})$ actually represents the overall solution quality for the induced sub-ILP from $\mathbf{d}$. When $\tau$ is small, its density would concentrate on the destroyed sets that contain the locally optimal solution. Therefore, we approximate $p_d$ with the predicted distribution over the locally optimal destroy sets with an adjustable variance

$$p_d(\mathbf{d}|\mathbf{x}^{(t)}) \approx \pi_\theta(\mathbf{d}|\mathbf{x}^{(t+1)}) = \prod_{i=1}^{n} \pi_\theta(\mathbf{d}_i|\mathbf{x}^{(t+1)}), \qquad (9)$$

where $\pi_\theta(\mathbf{d}_i|\mathbf{x}^{(t+1)})$ is a Bernoulli distribution with $p = \frac{1}{1+\exp(-\lambda_i^{(t)}/\sigma)}$. Here, $(\lambda_1^{(t)}, \cdots, \lambda_n^{(t)}) = f_\theta(\mathbf{x}^{(t+1)})$ is the output logits from the neural destroy policy $f_\theta$, while $\sigma > 0$ controls the variance of this distribution with a similar role to $\tau$. When $\sigma \to 0$, the densities of $\pi_\theta(\mathbf{d}|\mathbf{x}^{(t+1)})$ would also concentrate on the predicted locally optimal destroyed sets.

The above formulation allows us to reuse the existing training methods for the destroy policy, such as imitation learning (Sonnerat et al., 2021) and contrastive learning (Huang et al., 2023b). For simplicity and ease of data collection, we build SPL-LNS on top of the imitation learning method in this work. Compared with the greedy update strategy where $\mathbf{x}^{(t+1)} = \arg\max_{\mathbf{x}'} p_r(\mathbf{x}'|\mathbf{d}, \mathbf{x}^{(t)})$, our above analysis indicate that simply adding a sampling stage in the repair operator (and an optional rejection sampling) would transform LNS into an MCMC process efficiently converging to the target distribution with a theoretical guarantee. When we gradually anneal $\tau$ and $\sigma$ towards 0, SPL-LNS becomes a simulated annealing algorithm capable of escaping local optima with a high chance. The full sampling algorithm is summarized in Algorithm 1 with main innovations highlighted in red.

### 3.3 TRAINING SPL-LNS WITH HINDSIGHT RELABELING

Although Equation 9 offers flexibility in reusing the existing training methods, it is generally unclear what would be the best strategy to collect the training data for such kind of a sampling-enhanced LNS method. Since the sampling strategy significantly expands the support of the sample distribution generated by LNS, using LB to generate supervision over this extensive support becomes even more expensive. Therefore, we propose a hindsight relabeling strategy to generate supervision more efficiently with the neural LNS solver itself.

Specifically, we solve an ILP from the training set with our sampling-enhanced neural LNS solver to generate samples $\mathbf{x}^{(0)}, \mathbf{x}^{(1)}, \cdots$. Then for each sample $\mathbf{x}^{(t)}$, we find the best solution within its neighborhood from the generated samples

$$\bar{\mathbf{x}}^{(t)} = \arg\min_{\mathbf{x} \in \{\mathbf{x}^{(0)}, \mathbf{x}^{(1)}, \cdots\}, d_H(\mathbf{x}, \mathbf{x}^{(t)}) \leq \bar{\eta}^{(t)}} \mathbf{c}^\top \mathbf{x}, \qquad (10)$$

where $\bar{\eta}^{(t)}$ could be different from the neighborhood size $\eta^{(t)}$ used to generate samples. We can use $\mathcal{I}_t = \{i | \bar{\mathbf{x}}_i^{(t)} \neq \mathbf{x}_i^{(t)}\}$ as the supervision to train the neural destroy policy with the loss

$$L(\theta) = \sum_t (\sum_{i \in \mathcal{I}_t} \frac{1}{1 + \exp(-\lambda_i^{(t)})} + \sum_{i \notin \mathcal{I}_t} \frac{\exp(-\lambda_i^{(t)})}{1 + \exp(-\lambda_i^{(t)})}). \tag{11}$$

In our implementation, we first train a neural destroy policy using the samples collected by LB with the greedy update strategy. Then we use this neural LNS solver to collect more training samples with the above method. Finally, we combine the samples collected by LB and the neural solver to train another neural destroy policy for SPL-LNS.

# 4 EXPERIMENTS

## 4.1 EXPERIMENTAL SETUP

**Benchmark Datasets**  Following the previous evaluations in the literature (Huang et al., 2023b), we use 4 benchmark datasets for a variety of synthetic CO problems. Those problems include Minimum Vertex Covering (MVC), Maximum Independent Set (MIS), Combinatorial Auction (CA), and Set Covering (SC). We generate 200 and 50 small instances on each problem to collect the LB demonstrations for training and validation. An additional 100 small instances and 100 large instances are used for evaluation, where the large instances contain twice as many variables as the small instances. To distinguish between evaluations on instances of varying sizes, we employ the "-S" and "-L" suffixes to indicate small and large instances respectively. Besides, we also include one real-world dataset Load Balancing from the ML4CO competition (Gasse et al., 2022). We summarize all statistics for these problems in Table 1.

We use the same procedure in (Huang et al., 2023b) to generate all instances and training data. For LB, we use $\eta^{(0)}$ as 50, 500, 200, 50 and 200 for MVC, MIS, CA, SC and Load Balancing respectively, and $\gamma$ is fixed as 1 on all datasets. We use SCIP (version 8.0.1) to resolve ILPs formulated in LB and restrict the run-time limit to 60 minutes per iteration on each problem.

Table 1: Statistics for the problem instances in each dataset. The number of variables and constraints are reported for instances in each dataset.

| | Small Instances | | | | Large Instances | | | | |
|---|---|---|---|---|---|---|---|---|---|
| Dataset | MVC-S | MIS-S | CA-S | SC-S | MVC-L | MIS-L | CA-L | SC-L | Load Balancing |
| # Variables | 1,000 | 6,000 | 4,000 | 4,000 | 2,000 | 12,000 | 8,000 | 8,000 | 61,000 |
| # Constraints | 65,100 | 23,977 | 2,675 | 5,000 | 135,100 | 48,027 | 5,353 | 5,000 | 64,305 |

**Baselines**  We compare our methods to three non-neural baselines and two neural baselines featured by learning from LB, which include (1) BnB: the standard branch-and-bound algorithm used in SCIP (version 8.0.1), (2) RANDOM: an LNS algorithm selecting the neighborhood with uniform sampling without replacement (3) VARIABLE: an LNS algorithm with a variable neighborhood (Mladenović and Hansen, 1997) (4) LB-Relax: an LNS algorithm which selects the neighborhood with the LB-Relax heuristic (Huang et al., 2023a) (5) IL-LNS: a neural LNS algorithm which learns the LB heuristic through imitation learning (Sonnerat et al., 2021) and (6) CL-LNS: a recent neural LNS baseline which learns the LB heuristic via contrastive learning (Huang et al., 2023b). All of the LNS baselines adopt a greedy update strategy. On synthetic datasets, we train all neural methods on the demonstrations generated by LB on small instances and evaluate them on both small and large instances in the testing set. While on Load Balancing, the neural methods are trained on the training instances with a same distribution of the testing instances. This amounts to in total 9 testing datasets.

**Metrics**  We mainly compare the primal gap[1] and primal integral in this section while the results of the primal bound are presented in Appendix B.

---
[1]The optimal bound is calculated by the best result on each instance.

**Implementation Details** The training of SPL-LNS, IL-LNS and CL-LNS is kept the same across all datasets. We use 200 ILP training instances and 50 validation instances to generate the training data, while each ILP instance is represented as a bipartite graph (Gasse et al., 2019). A two-layer Graph Attention Network (Brody et al., 2022) is used as the backbone for all neural methods. The node features from the bipartite graph are first projected into a 64-dim embedding. We perform 2 rounds of message passing over the bipartite graph, with 8 attention heads and hidden size 64. We train each neural model for 30 epochs, using an AdamW optimizer with learning rate $10^{-3}$ and weight decay $5 \times 10^{-5}$. The batch size is set as 4.

The training of SPL-LNS+R is basically the same as in above neural methods, except with more training data collected by relabeling. To collect training data for SPL-LNS+R, we run SPL-LNS on all training and validation instances. The LNS starts with the best solution found by LB on each instance, and all other parameters of LNS are kept the same as the ones used for inference, except the running time on each instance is restricted to 30 minutes. For $t$-th intermediate solution, we check along the whole trajectory to find the best solution within its $\eta^{(t)}$ neighborhood and use it to create the supervision. $\eta^{(t)}$ is set as the hamming distance between the $t$-th and $(t + 1)$-th sample.

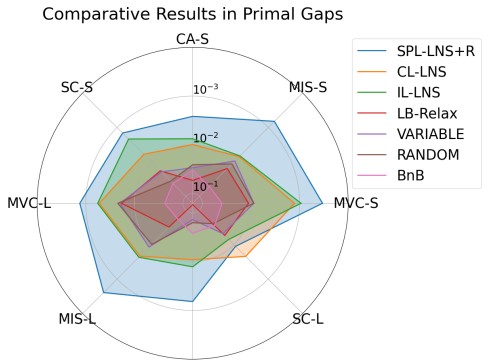

Figure 2: Comparative results in primal gaps at a 60-minute cutoff (lower value/outer ring is better). Note that the axes are reversed.

Figure 3: Comparative results in primal integrals at a 60-minute cutoff (lower value/outer ring is better). Note that the axes are reversed.

### 4.2 COMPARISON ON THE SYNTHETIC DATASETS

We evaluate all methods on the synthetic datasets for MVC, MIS, CA, and SC problems, following the hyperparameter settings in (Huang et al., 2023b). For VARIABLE, LB-Relax, IL-LNS, CL-LNS, and SPL-LNS, we set $\eta^{(0)}$ as 100, 3000, 1000, and 150 for MVC, MIS, CA, and SC respectively. For RANDOM, $\eta^{(0)}$ is set as 200, 3000, 1500, and 200 for MVC, MIS, CA, and SC. We fix $\gamma = 1.02$ and $\beta = 0.5$ in the adaptive neighborhood size for LNS-based methods across all datasets. During inference, all LNS methods start with the same feasible solution found by SCIP with a 30-second time limit. The run-time limit of SCIP on the sub-ILP $\mathbf{P}_t$ is restricted to 2 minutes.

Since the main contribution of this work is on the sampling part for the repair operator, we follow the same schedule of $\sigma$ as in Huang et al. (2023b), i.e., initially set as 0 and fixed as a constant after $\eta^{(t)} = \beta n$. For the schedule of the temperature $\tau$, we let $\tau^{(0)} = |\mathbf{c}^\top \mathbf{x}^{(0)}| + 1$ and iteratively decays it as $\tau^{(t+1)} = 0.9\tau^{(t)}$. The size of the candidate set $k$ is chosen from $\{3, 5\}$ based on the validation performance. We do not make careful tuning over the hyperparameters in SPL-LNS and some other choices may exhibit even better performance (e.g., see Section 4.4), but SPL-LNS has already demonstrated a clear advantage over our baselines which implies its general applicability. We train SPL-LNS via both the imitation learning and contrastive learning, which are denoted as SPL-LNS-IL and SPL-LNS-CL respectively. We continue to train SPL-LNS-IL with our hindsight relabelling method since SPL-LNS-IL achieves an overall better performance, and we denote this method SPL-LNS+R.

We first compare our complete sampling-enhanced LNS algorithm SPL-LNS+R with other baselines in Figure 2 and Figure 3, where the primal gaps and primal integrals at the 60-minute cutoff are presented. Please refer to Appendix B for detailed numbers (Table 3, 4, and 5). It can be seen

that SPL-LNS+R shows a significant advantage on all datasets except SC-L. One concern of the sampling method is the trade-off between the final performance (primal gap) and the convergence speed (primal integral), but due to the usage of the locally-informed proposal, SPL-LNS could also find a decent solution in a short time, without greatly sacrificing the primal integral.

To better understand the effects of the two proposed components, we visualize the dynamics of the primal gap on each dataset in Figure 4, including both SPL-LNS and SPL-LNS+R. we can see that the largest improvement against other baselines is still from the sampling strategy used in SPL-LNS, but the hindsight relabelling strategy could further improve the performance of SPL-LNS-IL.

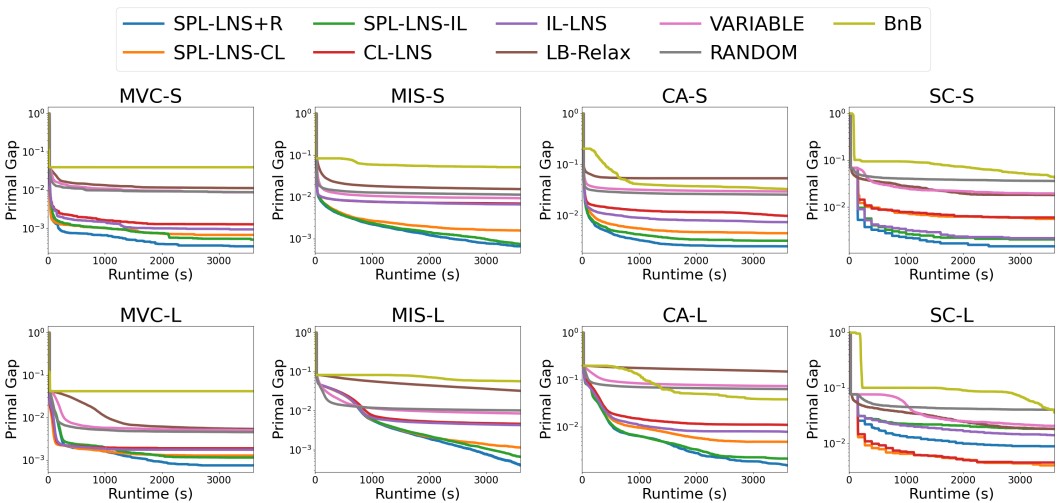

Figure 4: The plot of the primal gap (the lower is better) as a function of runtime on all datasets.

## 4.3 COMPARISON ON THE REAL-WOLD DATASET

We continue to compare our method with baselines on the real-world dataset, Loca Balancing. We utilize $\eta^{(0)} = 500$ for RANDOM and $\eta^{(0)} = 200$ for all other methods. Other hyperparameters are kept the same as in the last section. The comparative results are presented in Table 2 and Figure 5.

| Methods | BnB | RANDOM | VARIABLE | LB-Relax | IL-LNS | CL-LNS | SPL-LNS-IL | SPL-LNS-CL | SPL-LNS+R |
|---|---|---|---|---|---|---|---|---|---|
| Primal Gap (%) | **0.02** ± **0.06** | 0.10 ± 0.10 | 2.99 ± 1.13 | 1.45 ± 0.06 | 0.06 ± 0.09 | 1.26 ± 0.81 | 0.05 ± 0.08 | 1.28 ± 0.81 | 0.03 ± 0.06 |
| Primal Integral | 25.4 ± 14.1 | 23.3 ± 5.0 | 118.9 ± 40.6 | 63.8 ± 21.7 | 19.9 ± 3.5 | 69.9 ± 31.1 | 19.3 ± 3.2 | 71.4 ± 31.9 | **16.9** ± **3.4** |

Table 2: Comparative result of all methods in the **primal gap** (in %, lower is better) and primal integral (lower is better) at a 60-minute cutoff on the Load Balancing dataset. We bold the best result and color the second-best result in green.

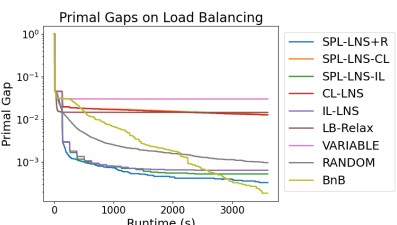

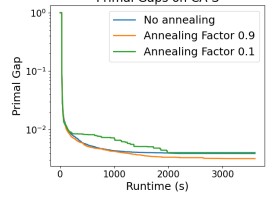

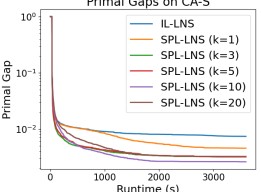

Figure 5: Comparative results on the Load Balancing dataset.

Figure 6: The ablation study on the effect of the annealing factor.

Figure 7: The ablation study on the effect of $k$.

Although BnB achieves a small advantage over all LNS methods in the final primal gap, it suffers from a poor primal integral. While SPL-LNS+R and SPL-LNS-IL beat all other LNS methods in the primal graph and take a clear lead in the primal integral over all methods. Therefore, we find our SPL-LNS method still effective on the real-world problems.

### 4.4 ABLATION STUDY

To further examine the effectiveness of our sampling method, we conduct ablation studies over main components of our SPL-LNS algorithm. We conduct all our ablation studies on the CA-S dataset.

We first investigate the sampling strategy in SPL-LNS, including the annealing factor and the effect of $k$. The usage of the top-$k$ sampling makes SPL-LNS method highly robust to the choice of the annealing factor, as we show in Figure 6. Basically, choosing any value that is reasonably large than 0, which corresponding to no sampling, would take effect. We continue to analyze the effect of $k$ by varying its value from $\{1, 3, 5, 10, 20\}$ and visualize the performance change of SPL-LNS in Figure 7. Remarkably, even $k = 3$ brings a significant improvement over the case where $k = 1$. This observation verifies the importance of sampling in escaping the "local optima". The performance reaches its best when $k = 10$ and drops back when $k = 20$, where a clear slowdown in the convergence speed can be observed. Such a trade-off in the exploration and exploitation is actually expected, but we can see that tuning $k$ for SPL-LNS is not hard and all values greater than 1 bring a significant improvement.

We then examine the effectiveness of our hindsight relabeling strategy. We fix the data collection time as 10 hours and use LB and hindsight relabeling method to collect the training data separately. We continue to train SPL-LNS-IL on the data collected by LB and name it as SPL-LNS+LB. We compare its performance with SPL-LNS+R in Figure 8. It can be seen that SPL-LNS+LB brings almost no improvement against SPL-LNS+IL due to its low collection efficiency, while SPL-LNS+R shows a clear improvement.

Although the greedy update strategy is widely used, some nuanced differences exist in the practical implementations. For example, Hottung and Tierney (2019) apply the rejection sampling on the best solution found by the repair operator, Song et al. (2020b) and Wu et al. (2021) do not consider the current solution during the greedy update, which corresponds to SPL-LNS with $k = 1$. All these methods allow a chance for the LNS to accept an inferior solution, which may alleviate the "local optima" problem to some extent. We compare SPL-LNS to these two update strategies with the same neural policy trained by IL-LNS. We denote these two update strategies as IL-LNS (rej) and SPL-LNS ($k = 1$) respectively. The comparative results are shown in Figure 9. It can be seen that both methods take effect in improving the performance of IL-LNS with a greedy update strategy, but their performance is still significantly worse than SPL-LNS-IL. We attribute this performance gap to the locally-informed proposal used by SPL-LNS, which balances the likelihood of the next candidate $p(\mathbf{x}')$ and the reverse proposal $q(\mathbf{x}^{(t)}|\mathbf{x}')$.

Finally, we verify whether the failure to find a better solution in the greedy update strategy is due to the short solving time of the repair operator. We extend the solving time of the ILP solver in IL-LNS from 2 minutes to 4 minutes and 8 minutes, and the results are presented in Figure 10. It can be seen that solving time has almost no effect on the final performance, which verifies our analysis in Section 3.1 that such a failure is due to the low chance to find a suitable destroyed set.

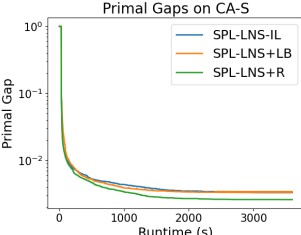

Figure 8: The ablation study on the hindsight relabeling.

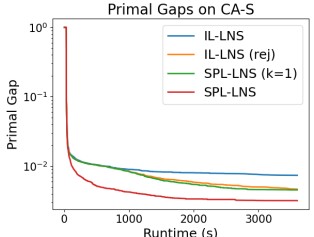

Figure 9: The ablation study on different update strategies.

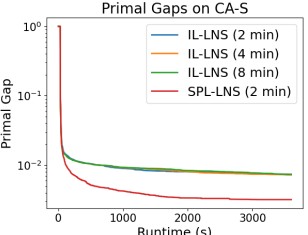

Figure 10: The ablation study on the effect of solving time.

## 5 RELATED WORK

### 5.1 LEARNING PRIMAL HEURISTIC FOR ILPs

The primal heuristics in ILPs aim to efficiently find high-quality feasible solutions. Diving and LNS are two main classes of primal heuristics and traditional solvers typically adopt a mixture of different variants of diving and LNS. Existing neural methods for primal heuristics mainly focus on heuristics selection (Khalil et al., 2017; Hendel et al., 2019; Chmiela et al., 2021), neural diving (Nair et al., 2020; Yoon, 2022; Han et al., 2023; Paulus and Krause, 2023) and neural LNS (Song et al., 2020a; Addanki et al., 2020; Sonnerat et al., 2021; Wu et al., 2021; Huang et al., 2023b).

LNS iteratively refines the solution by selecting a subset of decision variables (the neighborhood) to optimize at each time. Recent neural LNS methods mainly focus on the learning of neighborhood selection and leave the optimization to an off-the-shelf solver. Song et al. (2020a) partitioned the variables into subsets via a neural model and then searched over each neighborhood sequentially. Later, Wu et al. (2021) and Addanki et al. (2020) proposed more general frameworks for directly predicting the variables to optimize at each iteration. Recently, supervised-learning-based methods achieved state-of-the-art results by learning from the expert heuristic local branching through imitation learning (Sonnerat et al., 2021) or contrastive learning (Huang et al., 2023b). However, none of the existing methods have systematically studied the update rule in LNS. This is the first work to connect LNS with locally-informed proposals and propose a sampling-enhanced LNS method.

### 5.2 SAMPLING FOR COMBINATORIAL OPTIMIZATION

Sampling-based methods (Metropolis et al., 1953; Hastings, 1970; Neal, 1996; IBA, 2001) have been vastly applied in various CO problems (WANG et al., 2009; Bhattacharya et al., 2014; Tavakkoli-Moghaddam et al., 2007; Seçkiner and Kurt, 2007; Chen and Ke, 2004). However, previous methods typically suffered from a slow convergence speed compared with learning-based methods due to their inefficient proposals. The recent advances in MCMC have revitalized sampling-based methods and some of them have been successfully combined with neural models. Sun et al. (2023) first demonstrated that sampling-based methods could outperform neural CO solvers when using a locally-informed proposal. Sun and Yang (2023) and Li et al. (2023) successfully applied the diffusion models on CO problems, whose generative process is based on Langevin dynamics (Welling and Teh, 2011). However, the application of these sampling-based methods is greatly limited due to their incapability to generate feasible solutions directly. Instead, our proposed SPL-LNS can solve any CO problems formulated as ILPs and could be further extended to other non-linear or mixed programs when provided with an off-the-shelf solver.

## 6 CONCLUSION & LIMITATION

In this paper, we identify an important limitation of neural ILP solvers as the "local optima" issue. To address this concern, we propose SPL-LNS, a novel sampling-enhanced neural LNS solver demonstrating a strong capability in solving ILPs by escaping the "local optima". Specifically, we establish a theoretical understanding of the "local optima" problem in LNS, innovatively connect LNS and locally-informed proposals, and address the "local optima" problem with the simulated annealing approach. We also propose an efficient training strategy for this sampling-enhanced neural LNS solver. SPL-LNS is a generic framework that could be applied to various neural LNS solvers in combinatorial optimization problems, and we expect to extend it to other domains.

Although SPL-LNS is generally applicable and consistently shows improvement, it introduces additional design choices within the LNS framework that we have not systematically studied in this work. An effective LNS algorithm should balance the goal of finding a good solution quickly and continually improving the solution with extended search time. In this work, we adopt a relatively greedy strategy for the destroy operator, following previous research, but apply our annealing approach solely to the repair operator. However, exploring different combinations of these strategies could still yield valuable insights. Additionally, how to design a generic annealing strategy to apply neural solvers on heterogeneous ILP problems remains a significant challenge. We intend to explore this aspect in our future research.

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

## A DERIVATION OF THE LOCALLY-INFORMED PROPOSAL FOR LNS

Here we prove that the distributions in Equation 7 and 8 would give the locally-informed proposal defined in Equation 6. Before the formal proof, we shall note that for any feasible solution $\mathbf{x}'$ within the Hamming ball of radius $\eta^{(t)}$ centered around $\mathbf{x}^{(t)}$, there exists $\binom{n-d_H(\mathbf{x}',\mathbf{x}^{(t)})}{\eta^{(t)}-d_H(\mathbf{x}',\mathbf{x}^{(t)})}$ different choices of the destroy set with size $\eta^{(t)}$ such that the induced sub-ILP would include $\mathbf{x}'$ as its solution. Therefore, replacing $p_r$ and $p_d$ in Equation 4 with their formulations in Equation 7 and 8, we obtain

$$p(\mathbf{x}^{(t+1)}|\mathbf{x}^{(t)}) = \sum_{\|\mathbf{d}\|_1=\eta^{(t)}} p_r(\mathbf{x}^{(t+1)}|\mathbf{d},\mathbf{x}^{(t)})p_d(\mathbf{d}|\mathbf{x}^{(t)}) \tag{12}$$

$$= \sum_{\|\mathbf{d}\|_1=\eta^{(t)}} \frac{\exp\left(-\frac{1}{2\tau}E(\mathbf{x}^{(t+1)})\right)\mathbb{1}[\mathbf{x}^{(t+1)}\in\mathcal{S}(P(\mathbf{d},\mathbf{x}^{(t)}))]}{\sum_{\|\mathbf{d}'\|_1=\eta^{(t)}}\sum_{\mathbf{x}'\in\mathcal{S}(P(\mathbf{d}',\mathbf{x}^{(t)}))}\exp\left(-\frac{1}{2\tau}E(\mathbf{x}')\right)} \tag{13}$$

$$= \frac{\sum_{\|\mathbf{d}\|_1=\eta^{(t)}}\exp\left(-\frac{1}{2\tau}E(\mathbf{x}^{(t+1)})\right)\mathbb{1}[\mathbf{x}^{(t+1)}\in\mathcal{S}(P(\mathbf{d},\mathbf{x}^{(t)}))]}{\sum_{\|\mathbf{d}'\|_1=\eta^{(t)}}\sum_{\mathbf{x}'\in\mathcal{S}(P(\mathbf{d}',\mathbf{x}^{(t)}))}\exp\left(-\frac{1}{2\tau}E(\mathbf{x}')\right)} \tag{14}$$

$$= \frac{\sum_{\|\mathbf{d}\|_1=\eta^{(t)}}\exp\left(-\frac{1}{2\tau}E(\mathbf{x}^{(t+1)})\right)\mathbb{1}[\mathbf{x}^{(t+1)}\in\mathcal{S}(P(\mathbf{d},\mathbf{x}^{(t)}))]}{\sum_{\|\mathbf{d}'\|_1=\eta^{(t)}}\sum_{\mathbf{x}'\in\mathcal{S}(P(\mathbf{d}',\mathbf{x}^{(t)}))}\exp\left(-\frac{1}{2\tau}E(\mathbf{x}')\right)} \tag{15}$$

$$= \frac{\exp\left(-\frac{1}{2\tau}E(\mathbf{x}^{(t+1)})\right)\binom{n-d_H(\mathbf{x}^{(t+1)},\mathbf{x}^{(t)})}{\eta^{(t)}-d_H(\mathbf{x}^{(t+1)},\mathbf{x}^{(t)})}}{\sum_{d_H(\mathbf{x}',\mathbf{x}^{(t)})\leq\eta^{(t)}}\exp\left(-\frac{1}{2\tau}E(\mathbf{x}')\right)\binom{n-d_H(\mathbf{x}',\mathbf{x}^{(t)})}{\eta^{(t)}-d_H(\mathbf{x}',\mathbf{x}^{(t)})}}, \tag{16}$$

Here, the last line is exactly what we have in Equation 6.

To see why Equation 6 is a locally-informed proposal, let $g(y) = \sqrt{y}$ and $K(\mathbf{x}',\mathbf{x}) = \binom{n-d_H(\mathbf{x}',\mathbf{x}^{(t)})}{\eta^{(t)}-d_H(\mathbf{x}',\mathbf{x}^{(t)})}$ in Equation 2, then we could obtain Equation 6. Note that $d_H(\cdot,\cdot)$ is symmetric here and so is $K(\cdot,\cdot)$.

## B DETAILS ABOUT THE RESULTS

Here we present the detailed numbers in the primal bound, primal gap, and primal integral corresponding to the plots in Section 4.2. They are included in Table 3, 4, and 5 respectively.

Table 3: Comparative result of all methods in the **primal bound** (lower is better) at a 60-minute cutoff. We bold the best result and color the second-best result in green.

| Dataset | MVC-S | MIS-S | CA-S | SC-S |
|---|---|---|---|---|
| BnB | $460.8 \pm 9.8$ | $-2012.6 \pm 26.4$ | $-113374.3 \pm 1601.2$ | $176.8 \pm 13.5$ |
| RANDOM | $446.6 \pm 11.0$ | $-2098.6 \pm 12.3$ | $-114239.2 \pm 1792.4$ | $175.6 \pm 12.6$ |
| LB-Relax | $447.8 \pm 11.3$ | $-2089.8 \pm 12.6$ | $-110953.9 \pm 1725.6$ | $172.4 \pm 12.4$ |
| IL-LNS | $443.1 \pm 9.7$ | $-2108.7 \pm 11.8$ | $-116369.8 \pm 1535.9$ | $169.6 \pm 12.0$ |
| CL-LNS | $443.2 \pm 9.7$ | $-2108.0 \pm 11.6$ | $-116088.3 \pm 1545.8$ | $170.3 \pm 12.2$ |
| SPL-LNS-IL | $442.9 \pm 9.6$ | $-2121.3 \pm 12.0$ | $-116860.9 \pm 1476.1$ | $169.6 \pm 12.0$ |
| SPL-LNS-CL | $442.9 \pm 9.6$ | $-2120.2 \pm 11.9$ | $-116732.2 \pm 1430.1$ | $170.2 \pm 12.2$ |
| SPL-LNS+R | $\mathbf{442.8 \pm 9.7}$ | $\mathbf{-2121.4 \pm 11.5}$ | $\mathbf{-116944.4 \pm 1480.8}$ | $\mathbf{169.5 \pm 12.0}$ |

| | MVC-L | MIS-L | CA-L | SC-L |
|---|---|---|---|---|
| BnB | $920.0 \pm 12.3$ | $-3996.6 \pm 75.4$ | $-224196.4 \pm 2901.3$ | $111.6 \pm 7.3$ |
| RANDOM | $885.4 \pm 12.5$ | $-4194.2 \pm 16.5$ | $-218341.7 \pm 2694.0$ | $112.2 \pm 7.5$ |
| LB-Relax | $886.2 \pm 12.7$ | $-4100.4 \pm 21.2$ | $-198292.9 \pm 2602.1$ | $109.6 \pm 7.0$ |
| IL-LNS | $882.9 \pm 12.5$ | $-4219.2 \pm 16.2$ | $-231284.2 \pm 2623.3$ | $109.2 \pm 7.1$ |
| CL-LNS | $883.1 \pm 12.6$ | $-4217.8 \pm 16.1$ | $-230550.5 \pm 2547.5$ | $108.1 \pm 6.7$ |
| SPL-LNS-IL | $882.4 \pm 12.4$ | $-4234.6 \pm 15.4$ | $-232622.9 \pm 2386.4$ | $109.6 \pm 7.3$ |
| SPL-LNS-CL | $882.3 \pm 12.6$ | $-4234.4 \pm 14.1$ | $-232091.8 \pm 2543.0$ | $\mathbf{108.0 \pm 6.7}$ |
| SPL-LNS+R | $\mathbf{882.0 \pm 12.5}$ | $\mathbf{-4235.7 \pm 15.1}$ | $\mathbf{-232764.2 \pm 2484.6}$ | $108.6 \pm 7.0$ |

Table 4: Comparative result of all methods in the **primal gap** (in %, lower is better) at a 60-minute cutoff. We bold the best result and color the second-best result in green.

| Dataset | MVC-S | MIS-S | CA-S | SC-S |
|---|---|---|---|---|
| BnB | $3.94 \pm 0.52$ | $5.19 \pm 1.12$ | $3.29 \pm 0.58$ | $4.24 \pm 1.69$ |
| RANDOM | $0.88 \pm 1.16$ | $1.14 \pm 0.25$ | $2.55 \pm 0.93$ | $3.61 \pm 1.29$ |
| LB-Relax | $1.12 \pm 1.34$ | $1.55 \pm 0.26$ | $5.36 \pm 0.85$ | $1.81 \pm 1.07$ |
| IL-LNS | $0.09 \pm 0.13$ | $0.67 \pm 0.20$ | $0.74 \pm 0.41$ | $0.22 \pm 0.38$ |
| CL-LNS | $0.13 \pm 0.13$ | $0.70 \pm 0.19$ | $0.98 \pm 0.46$ | $0.60 \pm 0.71$ |
| SPL-LNS-IL | $0.05 \pm 0.08$ | $0.07 \pm 0.10$ | $0.32 \pm 0.28$ | $0.20 \pm 0.38$ |
| SPL-LNS-CL | $0.07 \pm 0.09$ | $0.16 \pm 0.12$ | $0.46 \pm 0.28$ | $0.56 \pm 0.69$ |
| SPL-LNS+R | $\mathbf{0.03 \pm 0.06}$ | $\mathbf{0.07 \pm 0.10}$ | $\mathbf{0.25 \pm 0.23}$ | $\mathbf{0.14 \pm 0.31}$ |
| | MVC-L | MIS-L | CA-L | SC-L |
| BnB | $4.19 \pm 0.41$ | $5.68 \pm 1.69$ | $3.83 \pm 0.74$ | $3.62 \pm 1.56$ |
| RANDOM | $0.46 \pm 0.24$ | $1.02 \pm 0.19$ | $6.34 \pm 0.65$ | $4.05 \pm 1.34$ |
| LB-Relax | $0.54 \pm 0.27$ | $3.23 \pm 0.29$ | $14.94 \pm 0.81$ | $1.84 \pm 1.14$ |
| IL-LNS | $0.18 \pm 0.20$ | $0.43 \pm 0.14$ | $0.79 \pm 0.48$ | $1.42 \pm 1.04$ |
| CL-LNS | $0.19 \pm 0.16$ | $0.46 \pm 0.13$ | $1.10 \pm 0.59$ | $0.46 \pm 0.72$ |
| SPL-LNS-IL | $0.12 \pm 0.19$ | $0.07 \pm 0.08$ | $0.21 \pm 0.27$ | $1.83 \pm 1.24$ |
| SPL-LNS-CL | $0.13 \pm 0.11$ | $0.11 \pm 0.08$ | $0.48 \pm 0.31$ | $\mathbf{0.40 \pm 0.61}$ |
| SPL-LNS+R | $\mathbf{0.08 \pm 0.12}$ | $\mathbf{0.04 \pm 0.07}$ | $\mathbf{0.15 \pm 0.23}$ | $0.89 \pm 0.77$ |

Table 5: Comparative result of all methods in the **primal integral** (lower is better) at a 60-minute cutoff. We bold the best result and color the second-best result in green.

| Dataset | MVC-S | MIS-S | CA-S | SC-S |
|---|---|---|---|---|
| BnB | $144.3 \pm 18.5$ | $220.7 \pm 38.5$ | $199.6 \pm 30.0$ | $344.7 \pm 82.0$ |
| RANDOM | $65.3 \pm 41.5$ | $75.9 \pm 8.7$ | $127.5 \pm 33.1$ | $173.3 \pm 46.4$ |
| LB-Relax | $77.4 \pm 46.2$ | $97.7 \pm 8.5$ | $225.3 \pm 30.9$ | $119.1 \pm 38.1$ |
| IL-LNS | $34.9 \pm 6.3$ | $57.5 \pm 7.1$ | $62.3 \pm 15.6$ | $49.6 \pm 14.2$ |
| CL-LNS | $36.1 \pm 4.9$ | $57.8 \pm 6.6$ | $74.6 \pm 17.8$ | $64.4 \pm 25.7$ |
| SPL-LNS-IL | $33.6 \pm 7.0$ | $38.5 \pm 3.5$ | $46.0 \pm 10.0$ | $48.9 \pm 13.7$ |
| SPL-LNS-CL | $33.5 \pm 3.4$ | $40.1 \pm 4.5$ | $51.5 \pm 10.0$ | $63.8 \pm 26.1$ |
| SPL-LNS+R | $\mathbf{32.4 \pm 3.5}$ | $\mathbf{37.9 \pm 3.7}$ | $\mathbf{43.1 \pm 8.0}$ | $\mathbf{46.2 \pm 11.6}$ |
| | MVC-L | MIS-L | CA-L | SC-L |
| BnB | $154.7 \pm 14.5$ | $261.4 \pm 28.6$ | $320.2 \pm 68.4$ | $498.7 \pm 76.7$ |
| RANDOM | $51.2 \pm 8.3$ | $76.8 \pm 6.8$ | $280.9 \pm 21.5$ | $194.5 \pm 47.9$ |
| LB-Relax | $72.9 \pm 8.6$ | $208.2 \pm 9.1$ | $634.2 \pm 27.4$ | $137.6 \pm 40.1$ |
| IL-LNS | $38.9 \pm 7.1$ | $69.0 \pm 4.7$ | $81.4 \pm 15.0$ | $105.3 \pm 37.1$ |
| CL-LNS | $39.1 \pm 5.6$ | $71.5 \pm 4.6$ | $95.8 \pm 17.0$ | $62.1 \pm 25.0$ |
| SPL-LNS-IL | $39.1 \pm 7.1$ | $61.9 \pm 3.7$ | $63.5 \pm 10.2$ | $115.4 \pm 41.9$ |
| SPL-LNS-CL | $37.2 \pm 4.0$ | $61.7 \pm 3.6$ | $75.7 \pm 11.4$ | $\mathbf{60.6 \pm 22.2}$ |
| SPL-LNS+R | $\mathbf{37.6 \pm 4.7}$ | $\mathbf{60.9 \pm 3.2}$ | $\mathbf{63.3 \pm 9.3}$ | $82.4 \pm 27.5$ |

