# OpenReview forum: "Sampling-Enhanced Large Neighborhood Search for Solving Integer Linear Programs"
_ICLR.cc/2025/Conference — ICLR 2025 Conference Withdrawn Submission_

### Official Review · Reviewer_D4PZ · 2024-11-01

**Soundness:** 3
**Presentation:** 2
**Contribution:** 2
**Rating:** 5
**Confidence:** 3

**Summary:**

This paper proposes a sampling-enhanced neural LNS solver that formulates the LNS as a stochastic process and a hindsight relabeling method to collect training data. Experimental results demonstrate the advantages of the proposed methods.

**Strengths:**

- This paper draws a connection between LNS and the MCMC with a locally-informed proposal.
- The hindsight relabeling trick can collect high-quality training data.

**Weaknesses:**

- I think $\eta$, $\tau$ and $\sigma$ are important parameters in this approach, and the author may want to conduct experiments on the effects of the different parameters.
- The presentation can be improved. The author could explain the advantages of the locally informed proposal and how it can help escape the local optima.
- Lack of necessary references on neural LNS, such as [1] and [2].

[1] GNN&GBDT-Guided Fast Optimizing Framework for Large-scale Integer Programming

[2] Light-MILPopt: Solving Large-scale Mixed   Integer Linear Programs with Lightweight  Optimizer and Small-scale Training Dataset

**Questions:**

- Could you please provide an analysis of why SPL-LNS is inferior to CL-LNS in the SC-L dataset?
- Could you please provide more insight on the better performance of BnB in the primal gap in the real-world dataset?

---

### Official Review · Reviewer_qxmP · 2024-11-04

**Soundness:** 3
**Presentation:** 2
**Contribution:** 1
**Rating:** 3
**Confidence:** 3

**Summary:**

This paper studies the large-neighborhood heuristics that work with an integer linear programming solver.

In particular, it proposes a locally informed method for sampling the next assignment (rather than greedy selection) and using simulated annealing to deal with local optima. The locally informed proposal, based on Zanella '17, connects LNS with discrete MCMC.

Experiments follow an existing dataset on Vertex Cover, Independent Set, Auctions, and Set Cover that compares the method with an LNS approach called Local Branching (and its two variants with imitation learning and contrastive learning), with default SCIP MIP solver, random LNS and variable LNS.

**Strengths:**

Formulation of LNS as a stochastic process, as in MCMC, is quite interesting.
Being stuck in local optima seems relevant to integer solving using LNS; hence, a study of this adds value to the literature.
LNS seems to benefit from sampling and simulated annealing per the experimental results. (see my comments on this below)
Making connections between LNS and locally-informed proposals is neat!

**Weaknesses:**

My main concern with this paper is that; at a high level, it is not clear what the paper is trying to achieve.

On the one hand, it starts with a broader question (and indeed a very interesting one!) about the local minimum behavior of LNS, which deserves its study.

On the other hand, it ends up tuning one particular LNS approach, a form of sampling augmented local branching, to perform better than its non-sampling versions and a few other baselines. If I understand it right, the original local branching method has already been shown to be better than the earlier approaches, so essentially, this paper shows that sampling improves over the results of local branching relaxation. This is nice, but rather a limited result compared to the paper's primary goal of "studying the local minima behavior of LNS".

Hence, there is a gap between what the paper proposes to study and what it presents/shows. If we treat the paper as a general study of local optima of LNS (which I find quite interesting!), unfortunately this is not what is experimented for. If we treat the paper as improving the best previous LNS, then it can be interpreted as relatively incremental. (plus, I am not sure if the extended training times lead to a fair comparison in the first place; see my comment below).

The theoretical framework is quite nice, and the intuition that the destroy operator finding a good set of variables for reaching a better solution becoming increasingly small seems reasonable/plausible.  What's not clear is how this is related to "neural LNS methods"? What makes this intuition specific to Neural methods only? Your proposal is more general than that, isn't it? Why is local optima only a problem for neural methods for local branching? Does local optima not pose an issue for other LNS destruction methods?

This paper would have been more impactful if it showed, within this neat theoretical model, that sampling and simulated annealing help several destroy operators (neural, non-neural etc.) to perform better than their original versions. In its current form, what is presented is sampling improves one particular LNS method. This is not to undermine the improvements provided by this combination. But in that case, we should not generalize this result too much into "studying the local optima behavior of LNS" and instead post it as a better LNS operator than previous work.

Alternatively, does your theoretical model suggest that other LNS destroy operators without neural network training do not exhibit this behavior? This is hard to believe, or at least not shown in the paper.

Other comments:

- The paper switches the terminology in a few places where sometimes Simulated Annealing is attributed to dealing with local minima, and sometimes Sampling is attributed to coping with local minima. Then, Figure 6 presents an ablation study in which top-k appears to be more effective than SA. If I understand correctly, while the k varies, the SA configuration is static. Could this not be a side effect of the particular hyper-parameterization of the SA? Why would SA prefer a fix/static k for different annealing schedules? If sampling K dominates SA, why is SA part of the overall method in the first place?

- The paper reads, "We could easily model Eq 8 using the feasible solutions found by the ILP solver". How does this work? What are the feasible solutions from ILP? Are we not solving the ILPs with an objective function? How do we obtain only satisfying/feasible assignments? Btw, if it's correct to generate feasible solutions by the ILP solver, than that means any other destroy operator can be augmented with some form of sampling strategy, as you did here for local branching, no?

- Have you considered comparing your sampling approach with some form of restart mechanism? Currently, none of the comparators in the experiments deal with the local optima as discussed here (other than some adaptive neighborhood size, if I understood that correctly, but they are not designed for dealing with local optima). So, I wonder if there would be better baselines (other than a list of different LNS) to better distinguish/address the sampling effectiveness. For instance, how stable is the initial feasible solution? Can the initial feasible solutions not be sampled and the process restarted, as a relatively simple baseline?

- Section 3.3 is hard to understand. Why would the training methodology differ for a sampling version? The goal of the training is to identify a good set of variables for the destruction operation, yes? Do you mean that for sampling, not only suitable destroy variables are needed but also a "diverse" set of suitable variables is required? Hence, more training? Also, if the sampling method is allowed for further training compared to other methods used in the comparisons, this is problematic, isn't it? How do we know the differences are not due to extended training here? Do you have results that compare sampling LNS with other LB LNS using the same amount of training budget?

Minor comments (that did not affect my review)
- when better solutions are far away from the current solution. What does that mean? What is far away? Do you mean objective-wise, or (hamming) distance in solution values, or else?
- The use of quotation marks is inconsistent. Please fix throughout
- typo Loca Balancing
- In parts, the paper refers to a concept called the "optimal destroy variable." What this means is not clear to me. What's an optimal destroy? Do you mean; a destroy that immediately leads to optimum value upon repair?

**Questions:**

- What's the main takeaway of this paper? To understand the local optima behavior of LNS or a sampling-inspired LNS that performs better than the previous best LNS?
- What makes your theoretical model specific to neural methods? This seems more general than that, no?
- Why is local optima only a problem for neural LNS?

---

### Official Review · Reviewer_vR3i · 2024-11-04

**Soundness:** 1
**Presentation:** 2
**Contribution:** 3
**Rating:** 3
**Confidence:** 5

**Summary:**

This paper proposed a refined LNS method SPL-LNS that help escaping local optima solutions. This method uses simulated annealing to sample next proposal based on not one but a group of feasible solutions. Also, a labeling technique is used to generate training data for SPL-LNS. The empirical results show strong performance of this method compared with other related methods.

**Strengths:**

1.	A sampling method transform LNS into an MCMC process. It helps escaping local optima by using simulated annealing algorithms.
2.	A hindsight relabel method is proposed to collect training samples instead of using expert collection rules in LB.

**Weaknesses:**

1.	The authors claim they identity the limitation of neural models in escaping local optima theoretically. But little evidence supports this claim.
2.	The main algorithms are displayed without any captions to explain the details and not much has been say in the main part of the paper regarding the accept and update functions.
3.	We all know simulated annealing method is very sensitive to the hyperparameter settings. As you decay by 0.9 every step in your experiments, have you tried other decay functions and parameters to test the robustness of your method?
4.	The experiments have been done in a relatively easy way. More realistic MIP datasets should be used like Item placement, mirplib library and even MIPLIB 2017.

**Questions:**

1.	As far as I know, most of your experiments have been done for binary problems, how would it perform if MIP problems are tested?
2.	Could you compare with more heuristic based LNS methods like RENS and RINS?
3.	Could you compare with two methods: Song et al. (2020b) and Wu et al. (2021)?

---

### Official Review · Reviewer_Lgrt · 2024-11-08

**Soundness:** 3
**Presentation:** 2
**Contribution:** 2
**Rating:** 5
**Confidence:** 4

**Summary:**

The paper introduces a Sampling-Enhanced Large Neighborhood Search (SPL-LNS) for solving Integer Linear Programs (ILPs) by addressing the limitations of neural network-based LNS methods in overcoming local optima. Traditional LNS methods often rely on a greedy approach, which can trap solutions in suboptimal neighborhoods. SPL-LNS proposes a stochastic reformulation that incorporates a locally-informed sampling strategy inspired by Markov Chain Monte Carlo (MCMC) and simulated annealing. Additionally, a hindsight relabeling strategy generates high-quality training data, allowing for a more robust destroy policy within the SPL-LNS framework. Empirical results demonstrate improvements over baseline methods on both synthetic and real-world ILP datasets.

**Strengths:**

Novel Sampling Strategy: By framing LNS as a stochastic process and introducing a locally-informed proposal distribution, the paper provides an innovative solution to the local optima issue, an ongoing challenge in combinatorial optimization.
Theoretical Foundation: The approach is supported by a solid theoretical foundation, detailing the MCMC and simulated annealing connections to LNS, and includes derivations that validate the efficacy of sampling.
Diverse Experiments: The experimental setup covers multiple synthetic ILP tasks (e.g., minimum vertex cover, combinatorial auction) and one real-world dataset from the ML4CO competition. This variety helps to demonstrate SPL-LNS’s robustness across different scenarios.

**Weaknesses:**

Limited Real-World Validation: The method has limited evaluation on real-world data, as the only non-synthetic dataset is from ML4CO, which is specifically designed for combinatorial optimization research. Additional real-world datasets from diverse applications could strengthen the paper's relevance.
Algorithmic Complexity: The proposed sampling strategy introduces additional computational complexity, especially with tuning parameters like temperature decay and the neighborhood size. The increased complexity may limit SPL-LNS’s practicality for larger-scale real-world applications.
Potential Overhead from Hindsight Relabeling: Although hindsight relabeling is intended to improve training data efficiency, it could add computational overhead to the training phase, particularly in scenarios with numerous variables and constraints.

**Questions:**

Scalability: How does SPL-LNS scale with very large ILP instances, particularly in terms of runtime and solution quality, when compared to traditional solvers?
Real-World Applicability: Can SPL-LNS generalize effectively to ILP problems that diverge from those tested here, such as scheduling, network design, or logistics, where problem structures and constraints might differ?
Parameter Sensitivity: How sensitive is SPL-LNS to parameter choices like the temperature decay rate, neighborhood size, and sampling rate? Could this sensitivity impact its usability in practical applications?

---

### Note · Authors · 2024-11-27

I have read and agree with the venue's withdrawal policy on behalf of myself and my co-authors.